# SEA: State-Exchange Attention for High-Fidelity Physics Based Transformers

**Parsa Esmati**
University of Bristol
parsa.esmati@bristol.ac.uk

**Amirhossein Dadashzadeh**
University of Bristol
a.dadashzadeh@bristol.ac.uk

**Vahid Goodarzi Ardakani**
Sabe Technology Limited
vahid.goodarzi@sabe.tech

**Nicolas Larrosa**
University of Bristol
nicolas.larrosa@bristol.ac.uk

**Nicolò Grilli**
University of Bristol
nicolo.grilli@bristol.ac.uk

## Abstract

Current approaches using sequential networks have shown promise in estimating field variables for dynamical systems, but they are often limited by high rollout errors. The unresolved issue of rollout error accumulation results in unreliable estimations as the network predicts further into the future, with each step's error compounding and leading to an increase in inaccuracy. Here, we introduce the State-Exchange Attention (SEA) module, a novel transformer-based module enabling information exchange between encoded fields through multi-head cross-attention. The cross-field multidirectional information exchange design enables all state variables in the system to exchange information with one another, capturing physical relationships and symmetries between fields. Additionally, we introduce an efficient ViT-like mesh autoencoder to generate spatially coherent mesh embeddings for a large number of meshing cells. The SEA integrated transformer demonstrates the state-of-the-art rollout error compared to other competitive baselines. Specifically, we outperform PbGMR-GMUS Transformer-RealNVP and GMR-GMUS Transformer, with a reduction in error of 88% and 91%, respectively. Furthermore, we demonstrate that the SEA module alone can reduce errors by 97% for state variables that are highly dependent on other states of the system. The repository for this work is available at: https://github.com/ParsaEsmati/SEA

## 1 Introduction

Solving partial differential equations (PDE) has been a primary concern of many fields in science and engineering, including physics [Salvini et al., 2024], chemistry [Grilli et al., 2020], and material sciences [Grilli et al., 2018, Esmati et al., 2024]. In many cases where a direct analytical solution of the PDE is impossible to obtain, numerical simulations are used. To solve these equations numerically, the domain is discretized into smaller cells using a discretization method such as finite element or finite volume methods. In some cases, the domain of interest is divided into millions of smaller elements forming large matrices representing the equation [Moukalled et al., 2016]. Solving these discretized equations typically follows an iterative technique, which can take days and sometimes weeks of runtime to converge to the full solution on the defined temporal domain [Liu et al., 2024, Jiang et al., 2023]. In some cases to resolve some specific features of the system a fine mesh is

38th Conference on Neural Information Processing Systems (NeurIPS 2024).

required. The graphical representation of small droplet [Um et al., 2018], cracking and brittle behaviour in thin components [Pfaff et al., 2014], multiphase and turbulent flows in computational fluid dynamics (CFD)[Kochkov et al., 2021, Thuerey et al., 2020, Heyse et al., 2021a,b] are all instances of such scenario.

Finer mesh however comes at the price of computational cost, which may not be feasible in industry settings. Consequently, there is a critical need for frameworks that can either bypass these detailed simulations or accelerate the solvers.

Recent advances in sequential networks have shown promising results in estimating the state variables of dynamical systems over time. However, they have not yet been effectively utilized to bypass or accelerate numerical models [Yousif et al., 2022]. Key challenges include lengthy training times and steep gradients in rollout errors [Sun et al., 2023, Han et al., 2022]. Additionally, many of these networks are task-specific, necessitating retraining for each unique test case [Li et al., 2020a]. If a model is to be integrated into a solver, steep error accumulation necessitates frequent retraining of networks to maintain efficacy when bypassing numerical solvers. This leads to unnecessary computational costs, making it challenging to integrate these models with solvers. Ideally, a model should be capable of learning the underlying physics and constitutive laws to minimize error gradients. Such a model could reduce retraining frequency and enable reuse through transfer learning approaches [Yosinski et al., 2014].

The current trend towards probabilistic models, such as diffusion models for science and physics, tends to overcome the rollout errors [Han et al., 2024, Valencia et al., 2024]. However, regardless of their domain (science, video, etc.), these models can produce unrealistic motions, especially for long sequence generations, and result in a temporal limitation [Weng et al., 2024]. Hence, the autoregressive generation seems unavoidable for longer sequences.

Transformer-based architectures, in particular, are at the forefront of the autoregressive generations [Sun et al., 2023, Zhao et al., 2023]. By drawing a parallel between the video and spatio-temporal PDE simulations, we intend to improve the unavoidable autoregressive rollout error for long sequence generation using transformers as the building block. Hence, inspired by the spatio-temporal cross-attention mechanisms used in vision models [Lin et al., 2022, Chen et al., 2021], we propose a State-Exchange Attention (SEA) module for physics-based transformers, designed to mimic PDE by explicitly capturing variable coupling. The effective multidirectional information exchange between fields enables the correction of some fields by others, allowing the model to learn the complex features of the physical system. This work employs a Vision Transformer (ViT) like [Dosovitskiy et al., 2020] encoder for mesh embedding. The embedded mesh at each instance is then used in the temporal State-Exchange Attention (SEA) integrated Transformer to predict the system's future states.

In summary, the **contributions** of this work include:

1. Design and integration of a novel SEA module for physics-domain Transformer models. This module enables learning the underlying physics through a multidirectional information exchange process between the state variables.

2. Assembly of a full ViT-SEA integrated framework that demonstrates state-of-the-art results in generating the complete sequence of the physical system given the initial sequence and specific time-invariant parameters representing the model.

3. Comprehensive evaluation of SEA module, and ViT-SEA integrated transformer across different computational fluid dynamics cases, showing over 60% reduction in error in all cases compared to state-of-the-art models.

## 2 Related work

In recent years, deep learning has led to major advancements in modeling physical systems, with contributions ranging from applying computer vision techniques to enhance the resolution of coarse meshes to incorporating physical symmetries and constraints through innovative modifications to learning objectives.

The work of [Raissi et al., 2019] demonstrated the feasibility of directly incorporating physical information into the objective function, including initial conditions and necessary physical residuals. Other examples of such approaches include [Jeon et al., 2024, Haghighat et al., 2021, Yu et al., 2022].

While these techniques embed physical knowledge into the objective function, they lack generality, as the obtained parameters tend to be case-specific. The broader underlying physics is not fully captured, and only the physical symmetries specific to a particular case are addressed.

Another class of methods is neural operators [Li et al., 2020b]. These methods generalize well across the PDEs they are trained on and are not case-specific. However, these models require data to be represented in higher dimensions to capture complex relationships. The application of an integral kernel in a high-dimensional representation of complex geometries with dense data points poses a significant computational challenge [Li et al., 2020a]. While neural operators can employ different architectures to reduce rollout error, they lack explicit mechanisms for integrating temporal data, unlike transformers, which handle sequence dependencies robustly through their attention mechanisms. This limitation can affect their effectiveness in applications that require sequential data processing.

A notable trend in recent years involves the use of encoder-decoder pairs to process dynamical states in latent space. The work of [Wiewel et al., 2019] is an early example of this approach. In general, the input must first be encoded into the latent space while preserving the context. A sequential network, such as LSTM, GRU, or other variants, is then trained on the encoded data. Thus, these approaches typically require two components: an encoder-decoder pair and a temporal model. Another example of this approach is Mesh Graph Networks (MGN) [Pfaff et al., 2020] and Graph-Network-based Simulators (GNS) [Sanchez-Gonzalez et al., 2020]. The encoder modules of these models are based on graph networks. After processing nodes, edges, and features, they use a simple multi-layer perceptron (MLP) to compute the derivatives of the features over time and update the state using a forward-Euler scheme. While the encoder-decoder pair is a powerful tool, especially for unstructured mesh spaces, the lack of a robust time-stepping model significantly limits performance, with rollout error becoming a dominant issue.

Current state-of-the-art models that demonstrate optimal performance on baseline datasets typically combine a graph network encoder with more advanced time-stepping algorithms. Notably, the study by [Han et al., 2022] employs a Graph Mesh Reducer (GMR) for encoding, along with a sequential time-stepping network and a Graph Mesh Up-Sampling (GMUS) decoder. This research explored various sequential models, including LSTM, GRU, and Transformers, with the latter achieving the lowest rollout error. Building on these principles, [Sun et al., 2023] introduced a modified version of GMR-GMUS, adding a RealNVP normalizing flow model to the time-stepping transformer. While adding a normalizing flow model does not yield a fully tractable model, it allows for the direct maximization of log-likelihood over the final data point, improving the overall objective and resulting in the most competitive baseline model reported thus far. However, these models still cannot fully address the rollout error in a systematic way that incorporates our physical understanding of the system.

As a result, improving rollout errors and reducing training time remain significant challenges in this field. In this work, we demonstrate that both objectives can be addressed by separating the fields and allowing them to learn the inherent physical relationships and symmetries. To this end, we introduce the SEA module, implemented on top of a Vision Transformer (ViT) based mesh autoencoder.

# 3 Methodology

## 3.1 Problem statement

Temporal generation of the states in a dynamical system from an initial condition is analogous to video generation tasks, where the process is conditioned on both the initial state and an external input. Similarly, the evolution of a dynamical system can be conditioned on known parameters, such as the Reynolds number in fluid flow and the system's initial condition. Given this similarity to autoregressive generative models like ART-V [Weng et al., 2024], we formulate the temporal evolution as an autoregressive generation of states, conditioned on both the initial state and a time-invariant parameter.

However, autoregression on the mesh space is challenging due to the large size of the elements and the number of variables stored on each element. Hence the mesh must be embedded into a manageable embedding, and our formulation becomes an autoregressive sequence generation in latent space.

If we denote the fields with index $i$, and time with index $t$, then the stored data on field $i$ at time $t$ is presented by $\mathbf{X}_t^i$. Given the proposed framework with the SEA module, we intend to encode the field groups separately. The field group refers to fields with the same dimensions (e.g., velocity in the $x$- and $y$-directions belong to the same group, while pressure lies in a separate group). Let $\mathcal{G}$ represent the partition of fields into groups based on their dimensions (e.g., velocity, pressure). Each group is then encoded separately with the mesh encoder $\mathcal{E}$, resulting in an encoded representation $\mathbf{Z}_t^{G_k}$ at time $t$. The final encoded representation at time $t$ is then given by:

$$\mathbf{Z}_t^{G_k} = \mathcal{E}\left(\text{Concat}\left(\{\mathbf{X}_t^i\}_{i \in G_k}\right)\right), \quad \forall G_k \in \mathcal{G}$$

We now denote $\mathbf{Z}_t^j$ as the encoded representation of the group $G_k$, where $j$ indexes the group embedding space. This embedding is then used to formulate the autoregressive sequence generation in time where the initial condition $\mathbf{Z}_0^j$ and the time invariant parameter $\Theta$ (e.g., Reynolds number) are always available and used to condition the generation. We can formally express this problem as $\arg\max_{\mathbf{Z}_{1:T}^j} P(\mathbf{Z}_{1:T}^j \mid \mathbf{Z}_0^j, \Theta)$, where we aim to find the sequence of latent variables $\mathbf{Z}_{1:T}^j$ that maximizes the conditional probability given the initial condition $\mathbf{Z}_0^j$ and the time-invariant parameter $\Theta$. This is achieved through a pointwise estimation of the conditional probabilities in the continuous embedding space, with optimization performed by minimizing the L2 loss. Further detail on objectives and training is provided in Appendix C.

### 3.2  ViT mesh autoencoder

The backbones commonly used to create embedding spaces in image and video models, such as Latent Diffusion Models (LDM) [Rombach et al., 2022] and Video Diffusion Models (VDM) [Ho et al., 2022], cannot be directly applied to mesh data due to their inherent structured pixel inductive biases. Therefore, our proposed autoencoder must specifically overcome these biases for unstructured mesh space. Given that the temporal model employs a transformer backbone, the embedding must generate tokens compatible with the transformer's input requirements. These tokens are generated similarly to that of ViT [Dosovitskiy et al., 2020]. Following the approach used in ViT, the space is divided into multiple patches, with each patch containing a number of cells. Let $\Omega$ denote the domain in which our study is conducted, with dimension $d$. Assume that the domain is discretized into a set of nodes $\{x_i\}_{i=1}^N$, where each node $x_i$ represents a point in $\mathbb{R}^d$. To construct patches, we define a partitioning function $f_P : \mathbb{R}^d \to \{1, 2, \ldots, M\}$, which assigns each node $x_i$ to one of $M$ patches based on its coordinates.

Assuming the boundaries between patches are showing with $B = \{b_1, b_2, \ldots, b_m\}$ then the function $f_P$ is defined as follows:

$$f_P(x_i) = \begin{cases} 1 & \text{if } x_i < b_1, \\ j & \text{if } b_{j-1} \leq x_i < b_j \quad \text{for } 2 \leq j \leq m, \\ m+1 & \text{if } x_i \geq b_m. \end{cases}$$

To address the challenge of irregular and unstructured meshes, which lead to varying numbers of nodes per patch, each patch is padded to align with the length of the largest patch. A padding value of 0 is used throughout the framework. Moreover, bias terms are excluded in the embedding process, and the Gaussian Error Linear Unit (GELU) activation function [Hendrycks and Gimpel, 2016] is applied to ensure that the padded elements do not influence the spatial encoding. To achieve spatially aware embeddings and coherent reconstructions, we apply a multi-head self-attention mechanism (MHSA). This padding and embedding strategy is illustrated in Figure 1.

The output generated by the Vision Transformer (ViT) embedding module is subsequently flattened and utilized as tokens within the temporal model. The complete process is thoroughly explained in appendix A.

### 3.3  Temporal and State-Exchange Attention model

The State-Exchange Attention module is integrated into the temporal model in this work to enhance the autoregressive generation of sequences in time. The temporal model utilizes a transformer

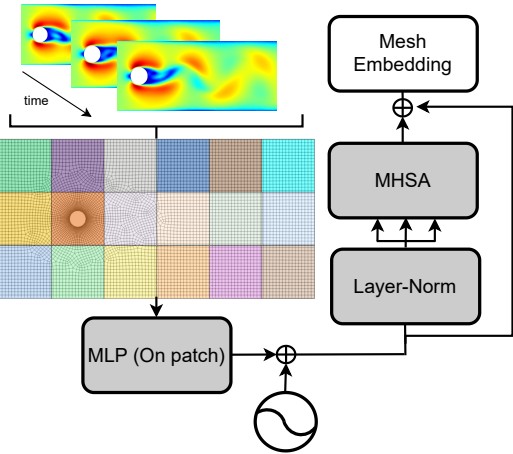

Figure 1: The ViT-based mesh autoencoder divides the domain into patches and pads them to ensure equal sizes. An MLP is then applied to reduce the dimensionality, and the MHSA mechanism provides global awareness to create spatially coherent reconstructions with minimal patch artifacts. These patches are subsequently flattened and adapted as tokens for the temporal model.

architecture to capture the temporal dependencies of the state variables. This transformer includes the adaptive layer-norm by [Peebles and Xie, 2023] and the rotational positional embedding (RoPE) as developed in [Su et al., 2024] and adapted by state of the art autoregressive image generation models [Lu et al., 2024, Sun et al., 2024]. The adaptive layer-norm employed in this work is modified to take the continuous time invariant parameters as input and act as a secondary conditioning mechanism on the temporal model. Full detail of the temporal model is provided in appendix B.

This work initializes a decoder block for each state variable in the given PDE For instance, the Navier-Stokes equations governing the fluid dynamics, requires the resolution of variables such as velocity and pressure each of these are assigned an expert decoder. The SEA module is designed to allow the exchange of information amongst these experts with cross attention. We further investigate other modes of information exchange in appendix D.

The flow of information through the expert layers and the SEA module can be formulated by building on the well-known attention mechanism. For clarity, the terms regarding the positional embedding are omitted here. We start from the encoded groups of variables presented in section 3.1, denoted by $\mathbf{Z}_t^j$. To simplify the notation and remove the explicit dependence on $t$, we represent the sequence of encoded variables across all time steps as a single matrix, $\mathbf{Z}^j$, which stacks the encodings of all time steps. The attention mechanism then reads:

$$\text{Attention}(\mathbf{Z}^j) = \text{softmax}\left(\frac{Q(\mathbf{Z}^j)K(\mathbf{Z}^j)^T}{\sqrt{d_k}}\right)V(\mathbf{Z}^j) \tag{1}$$

Where $K(\mathbf{Z}^j)$, $Q(\mathbf{Z}^j)$, and $V(\mathbf{Z}^j)$ represent the key, query, and value matrices, and are obtained from the linear transformation of the input $\mathbf{Z}^j$ by a set of learnable weights $\mathbf{W}_K$, $\mathbf{W}_Q$, and $\mathbf{W}_V$. Additionally $d_k$ represents the model dimension.

Furthermore the adaptive layer-norm is demonstrated by 'AdaLN' and hence following a pre-norm convention the multihead self-attention (MHSA) becomes:

$$(\mathbf{Z}^j)_{\text{SA}} = \mathbf{Z}^j + \text{Attention}\left(\text{AdaLN}(\mathbf{Z}^j)\right), \quad \forall j \in \mathcal{G} \tag{2}$$

Given the autoregressive task at hand all the attention mechanisms including the presented MHSA have causal mask to improve autoregressive generation.

The field information flows through the information exchange module after the temporal self attention. This information exchange module is represented by the state-exchange attention in Figure 2, where we allow the state variables to exchange relevant information with a causal cross attention mechanism. This is formulated as:

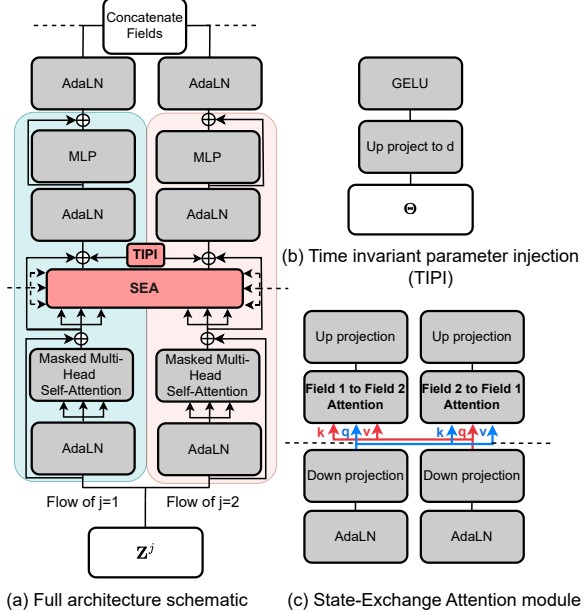

Figure 2: (a) State-Exchange Attention (SEA) integrated Transformer model architecture, incorporating the additional modules of SEA and Time Invariant Parameter Injection (TIPI). Dashed lines represent the inclusion of additional fields. (b) Representation of the TIPI, designed to incorporate time-invariant information after the SEA module. (c) Schematic of the SEA module, illustrating how fields communicate through this module.

$$(\mathbf{Z}^j)_{\text{SEA}} = (\mathbf{Z}^j)_{\text{SA}} + \sum_{\substack{k \in \mathcal{G} \\ k \neq j}} \mathrm{f}_{\text{SEA}}\left(\text{AdaLN}((\mathbf{Z}^j)_{\text{SA}}), \text{AdaLN}((\mathbf{Z}^k)_{\text{SA}})\right), \quad \forall j \in \mathcal{G} \qquad (3)$$

Here, $\mathrm{f}_{\text{SEA}}(\cdot, \cdot)$ represents our information exchange mechanism SEA. This module takes in two arguments: the first is the adaptive layer norm of the expert for which the attention is taking place, and the second is the adaptive layer norm from the expert to which the module is attending. These are represented by $\text{AdaLN}(\mathbf{Z}^j)$ and $\text{AdaLN}(\mathbf{Z}^k)$, where $k$ and $j$ are non-equal embedding field indices.

To enhance efficiency during information exchange, we adopt a bottleneck mechanism inspired by expert-based architectures such as [Lee et al., 2024]. Introducing a bottleneck at this stage helps keep the model scalable by enabling selective information exchange in a lower-dimensional space. Another instance of such strategy is the Perceiver architecture [Jaegle et al., 2021], where cross-attention creates a bottleneck for high-dimensional data from different modalities. In our case, we employ a simpler method, using a down-projection with learnable parameters, following the approach in [Lee et al., 2024]. If the down- and up-projection matrices for mapping to the bottleneck are denoted by $\mathbf{W}_d^j$ and $\mathbf{W}_u^j$, respectively, the State-Exchange Attention mechanism is fully described by Equation 4. The arguments to this function are the AdaLN of $(\mathbf{Z}^j)_{\text{SA}}$ and $(\mathbf{Z}^k)_{\text{SA}}$, as shown in Equation 3. However, for clarity in illustrating the equation, we use $\mathbf{A}^j$ and $\mathbf{A}^k$ to represent the input arguments here.

$$\mathrm{f}_{\text{SEA}}(\mathbf{A}^j, \mathbf{A}^k) = \mathbf{W}_u^k \left( \text{softmax}\left( \frac{Q(\mathbf{W}_d^j \mathbf{A}^j) K(\mathbf{W}_d^k \mathbf{A}^k)^T}{\sqrt{d_k}} \right) V(\mathbf{W}_d^k \mathbf{A}^k) \right) \qquad (4)$$

In the presented equation $K(\mathbf{W}_d^j \mathbf{A}^j)$, $Q(\mathbf{W}_d^j \mathbf{A}^j)$, and $V(\mathbf{W}_d^j \mathbf{A}^j)$ represent the key, query and value matrices obtained form the down projection of the inputs.

The conditioning of the generations on external parameters such as the Reynolds number is done using an indirect method of adaptive layer norm and a direct method of time invariant parameter injection module (TIPI). These components replace the more computationally intensive attention mechanism commonly used in physics domain autoregressive models. The TIPI component processes the time-invariant parameters using a multilayer perceptron (MLP) with a GELU activation function. The MLP maps the time-invariant parameters $\Theta$ to the model's embedding dimension through learnable parameters. As shown in Equation 5, the information injector, represented by $\text{TIPI}[\cdot, \cdot]$, injects these processed parameters into the current state by summing the model's embedded information with the upscaled time-invariant parameters.

$$(\mathbf{Z}_i^k)_{\text{Output}} = (\mathbf{Z}_i^k)_{\text{SEA}} + \text{MLP}\left(\text{AdaLN}\left(\text{TIPI}\left[(\mathbf{Z}_i^k)_{\text{SEA}}, \Theta\right]\right)\right) \tag{5}$$

Figure 2 illustrates the architecture's detailed schematic.

## 4 Experiments

### 4.1 Procedure

In this section, we evaluate the proposed model on two benchmark datasets and compare its performance with other frameworks. First, the complete model is tested on the cylinder flow benchmark, a widely used dataset in computational fluid dynamics. This evaluation includes the comparison of the full model with recent physics domain autoregressive models.

We then explore a multiphase case, where the model is tasked to resolve the interface by taking into account the fluxes caused by the velocities using the SEA module. In this section, only the temporal aspects of the model are varied (SEA module), while the ViT mesh autoencoder is fixed to isolate and eliminate the impact of encoding method on model's performance. To this end, separate decoders are assigned, one for velocity and another for volume fraction, similar to structure depicted in 2. The inclusion of the volume fraction allows us to study the extent of the model's capability to resolve interfaces and the effect of State-Exchange Attention on capturing the multiphase scenarios.

For consistent comparison with state-of-the-art models [Sun et al., 2023, Han et al., 2022], relative mean squared error is used to quantify the errors. The model was trained on an A100 GPU for approximately 2 hours for both datasets. Furthermore, a consistent Transformer architecture was adopted in both cases, utilizing 1 layer and 8 attention heads. The embedding dimension of the model for the cylinder flow case was set to 1024, in line with the literature [Sun et al., 2023], while a dimension of 2048 was used for the multiphase flow case to effectively capture the interface. Full details of the configurations and datasets are provided in Appendix G and F, respectively.

### 4.2 Evaluation of ViT mesh autoencoder

The autoencoder used in the following experiments was trained exclusively with a reconstruction objective. The complete training procedure for this model is detailed in Appendix A. Reconstruction errors, compared to recent graph-based autoencoders, are presented in Table 1.

Table 1: Relative reconstruction error for cylinder flow and multiphase flow reported in the unit of $1 \times 10^{-3}$.

| Dataset | Ours (ViT based autoencoder) | GMR-GMUS | PbGMR-GMUS |
|---|---|---|---|
| Cylinder flow | **1.7** | 14.3 | 1.9 |
| Multiphase flow | 6.3 | - | - |

### 4.3 Evaluation of temporal model on general case

We assess our complete architecture using the 2D cylinder flow, a benchmark dataset employed by other leading baseline models. In this case, the Navier-Stokes equation governs the motion of the fluid throughout the domain. Consequently, the trajectory to track is the velocity and pressure. The

mesh was initially tokenized at all time steps based on the explained ViT mesh encoder. These tokens were then fed through the SEA integrated model illustrated in Figure 2. During the training, the entire sequence was fed to the network for each batch. During inference, the model was set to estimate the trajectory autoregressively, and hence, the test errors correspond to the total rollout error. For the cylinder flow dataset, the error was evaluated over the case with Reynold's number of 400. This case was chosen to keep consistent with the other competitive models.

The recorded rollout error, and its comparison with other baseline models is illustrated in Figure 3.

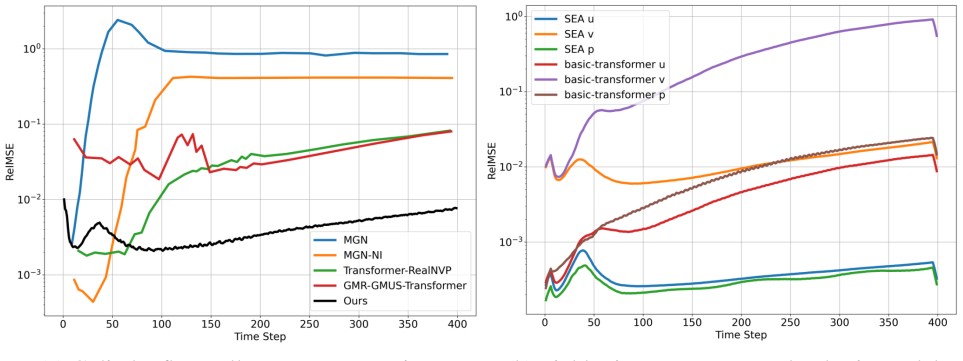

(a) Cylinder flow rollout error comparison          (b) Field-wise error compared to basic model

Figure 3: Comparison of the rollout error for the cylinder flow dataset.

In Figure 3a, we show that our model outperforms the established state-of-the-art models across the board. Specifically, our results indicate an average improvement of 88% and 91% in reducing the error in autoregressive sequence generation compared to the PbGMR-GMUS Transformer-RealNVP and GMR-GMUS Transformer architectures, respectively. Furthermore, our model outperforms both variants of MGN architecture with 99% and 98% improvement over the base MGN model and MGN-NI, respectively. Full detail of the errors are presented in Table 2.

Table 2: Time average rollout error of the presented model compared to other competitive models. The presented error is after decoding and corresponds to the real field error. The reported values are in the unit of $1 \times 10^{-3}$

| Models | u | v | p | **Avg** |
|---|---|---|---|---|
| MGN [Pfaff et al., 2020] | 98 | 2036 | 673 | 935.6 |
| MGN-NI [Pfaff et al., 2020] | 25 | 778 | 136 | 313 |
| GMR-GMUS Transformer [Han et al., 2022] | 4.9 | 89 | 38 | 43.96 |
| PbGMR-GMUS Transformer-RealNVP [Sun et al., 2023] | 3.8 | 74 | 20 | 32.6 |
| Ours (Full ViT-SEA Transformer) | **0.35** | **10.7** | **0.3** | **3.7** |

The evaluated test cases were post-processed to generate a contour map for further observation of the learned patterns and potential areas of error. For consistency with the work in literature [Han et al., 2022, Sun et al., 2023] the contour map of the case with Reynolds number of 400 is presented here. To fully explore our model's ability to capture complex flow features, such as the downstream vortex, we present visualizations at the 250th timestep, a point at which the Von Karman vortex street is fully developed, as shown in Figure 4. This timestep was deliberately chosen to visualize complex dynamics that are often challenging for traditional models to capture.

## 4.4   Evaluation of temporal model on multi-phase

Evaluation of the presented model is extended to include a multiphase flow scenario, which complements the analysis presented in Section 4.3. The evaluated test case in this experiment corresponds to an immiscible collapse of two blocks of liquid due to density differential. In studies of multiphase flows, a critical aspect is the precise identification of fluid interfaces. This is accomplished by using a volume fraction state variable, denoted by $\alpha$, which indicates the region occupied by the fluid. The value of $\alpha$ ranges from 0 in one fluid to 1 in the other, effectively distinguishing between the

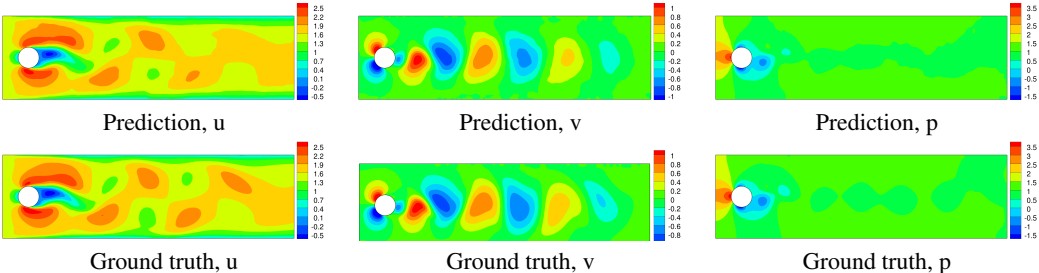

Figure 4: Contour maps of the generated fields at Re=400, and time step of 250 where Von Karman vortex street is formed.

two phases. To capture the phase, an additional block is assigned to the volume fraction which can communicate with other fields (Velocity in this case). Given the minor variations of pressure, we discard this variable here and only focus on the importance of the field communication between velocities and volume fraction through SEA module.

We investigate three possible variations of the Transformers to achieve this. First, we estimate the rollout error of the model with the SEA module. Second, we evaluate a basic model that encodes the fields into different latent spaces with no mode of information exchange. Finally, we assess a model that encodes all fields together into a single latent space, referred to as the Field Fusion Encoder (FFE) Transformer. This latter model corresponds to the Transformer architecture used in the PbGMR-GMUS Transformer-RealNVP and GMR-GMUS Transformer, as indicated in the provided results. The rollout error of these models are presented in Figure 5.

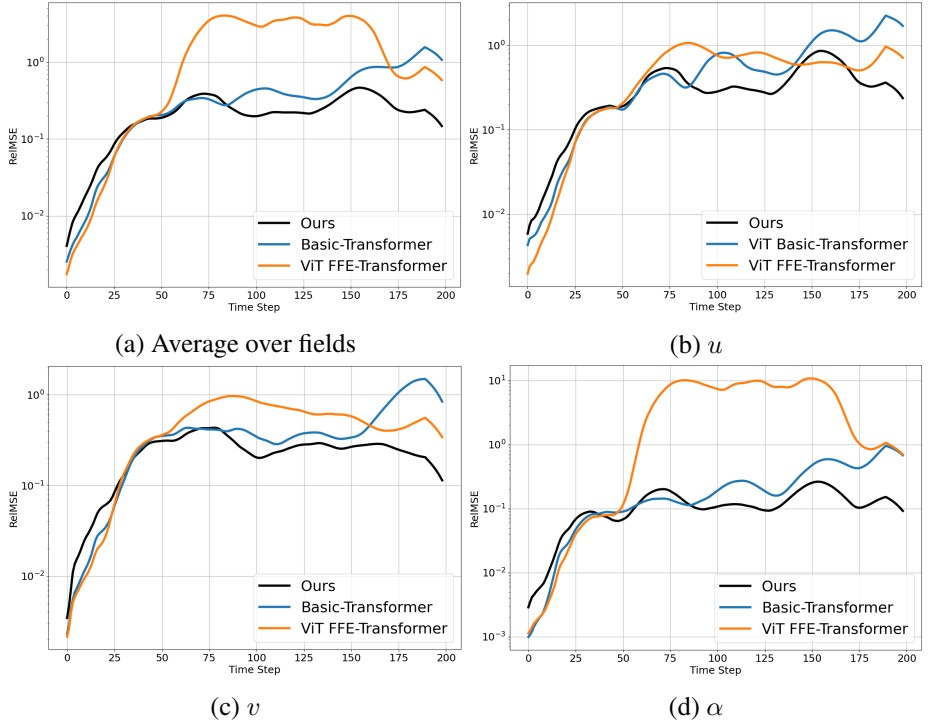

Figure 5: Comparison between the Transformer with SEA module and other variations of Transformers used in the literature over the multi-phase dataset.

From the comparative results presented in Figure 5, it is evident that the models with the SEA module outperform the other variations. Most of the error observed in the average error plot in Figure 5 corresponds to the error in the volume fraction. The mean volume fraction errors over all time steps are 0.12, 0.25, and 4.61 for the SEA-integrated Transformer, basic Transformer, and FFE Transformer,

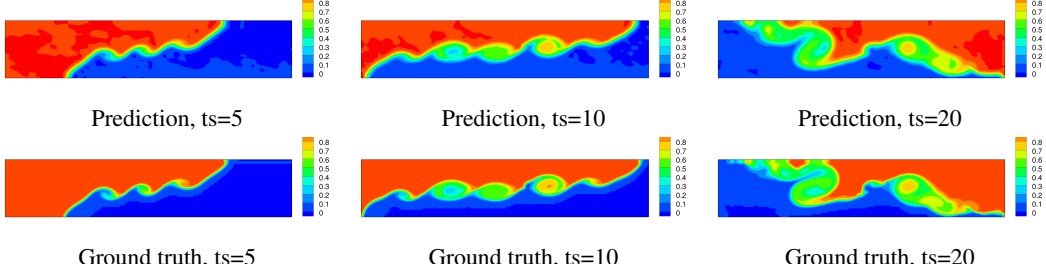

Figure 6: Comparison of the contour maps of the volume fraction ($\alpha$), between the predictions and ground truth in the case of $\rho = 850$

respectively. This represents approximately a 52% and 97% reduction in error with the integration of the SEA module, compared to the basic and FFE Transformers. Additionally, improvements of 48.5% and 40% are observed in the averaged velocity components.

Further demonstration of the contour maps is provided in Figure 6 where the actual interface tracking capability of the SEA enhanced transformer can be observed. Further visual results on the velocities are provided in Appendix H.

## 5   Discussion

The presented ViT-based mesh autoencoder and State-Exchange Attention (SEA) integrated Transformer module were evaluated through two different experiments on computational fluid dynamics (CFD) problems. A significant improvement in relative mean squared error was observed when the SEA module was deployed. During the cylinder flow evaluation, the full ViT-SEA integrated transformer framework achieved over 80% improvement compared to all competitive baselines. This improvement was accompanied by a lower gradient in the error, demonstrating a form of self-correction through information exchange between the velocity and pressure fields. The isolated SEA module was then tested on a multiphase case, resulting in a 97% improvement in the volume fraction compared to the field fusion encoder transformer, where the entire field is encoded into the same latent space. In the multiphase case, it was evident that the velocities exhibited a marginal error difference; approximately 40-50%; however, the volume fraction dominated the overall improvement. This is due to the significance of velocity in the displacement of the interface, whereas the volume fraction did not provide any valuable information to the velocity field. The error reduction observed with the deployment of the SEA module suggests that SEA module provides the necessary tools for the underlying physics of the governing to be captured.

The presented module, however, may face challenges when scaling to equations involving a large number of state variables. For instance, in multiphase flows with more than two phases or in grain growth within materials where each grain is represented by a state variable, the model would require a corresponding number of transformers to operate in parallel. This could lead to inefficiencies as the number of variables increases significantly.

## 6   Conclusion

The presented work introduces SEA, a novel module that enables the state variables of a physical system to exchange information within the transformer architecture. Transformers integrated with SEA demonstrated state-of-the-art performance, surpassing previously established benchmarks by other transformer-based models with improved autoregressive rollout error. The improved rollout error is an indication that the SEA module is enabling the model to learn the underlying physics. In future works, we will explore the scaling complexity of SEA to handle more state variables and its application in other domains. Furthermore, the integration of SEA with probabilistic models, such as diffusion models, will be investigated.

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

# Appendix

## A    ViT autoencoder for Mesh detail

Embedding mesh data presents a greater challenge compared to images, where a structured pixel grid allows for the natural application of convolutional kernels. The unstructured nature of meshes requires novel approaches for embedding, such as graph neural networks [Pfaff et al., 2020], or systematic methods for padding within the kernels. However, determining the appropriate padding and their position in the convolution is not a simple task.

ViT models, originally developed for image classification, partition images into patches and apply a linear projection to each patch. Inspired by this structure, we developed a ViT-like mesh autoencoder, where patches are generated on the mesh, and each patch is encoded individually. In this approach, the number of cells per patch becomes readily available, allowing all patches to be padded to the length of the patch with the maximum number of cells. To enhance the processing capability during the embedding stage, we replace the original patch projection in the ViT with an MLP layer. The self-attention block applied after this step further refines the embedding, creating spatial coherence by allowing each patch to capture the global context in addition to the local context encoded by the MLP.

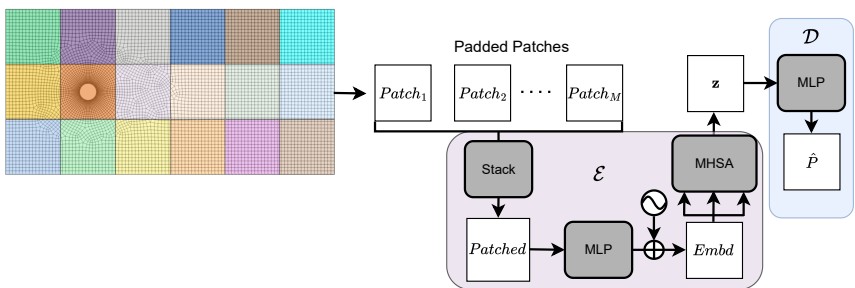

Figure 7: Presentation of the ViT mesh autoencoder, where a number of patches are created on the mesh, and an embedding space is created for each patch using the encoder $\mathcal{E}$. The multihead self attention is then applied to create a global awareness. The decoder $\mathcal{D}$ simply maps the embedded patches to the original space.

The following list outlines the evolution of the data structure as it passes through the ViT-based mesh autoencoder. Here, $B$ denotes the batch size, $T$ represents the number of time steps, $C$ is the total number of mesh cells, $F$ refers to the number of fields (analogous to channels in image or video contexts), $P$ is the number of patches, $C_p$ is the number of cells per patch, and $D$ is the embedding dimension. In this study, batches are derived from the generated trajectories, as described in Section F.

Input:    $[B, T, C, F]$
Patchify ($Patched$):    $[B, T, C, F] \rightarrow [B, T, P, C_p, F]$
Permute:    $[B, T, P, C_p, F] \rightarrow [B, T, P, F, C_p]$
Collapse batch with time and field with cell:    $[B * T, P, F, C_p] \rightarrow [B * T, P, F * C_p]$
Encode with MLP ($Embd$):    $[B * T, P, F * C_p] \rightarrow [B * T, P, D]$
Global awareness with MHSA (**z**):    $[B * T, P, D] \rightarrow [B * T, P, D]$
Decode with MLP ($\hat{P}$):    $[B * T, P, D] \rightarrow [B * T, P, F * C_p]$

The presented procedure is done for fields within each field group separately. The field groups refer to the groups of fields with the same physical dimension as formalised in Section 3.1.

For completeness, we also provide a variational version of the encoder in the repository, incorporating a weighted KL divergence penalty, following the approach in [Rombach et al., 2022]. However, in our

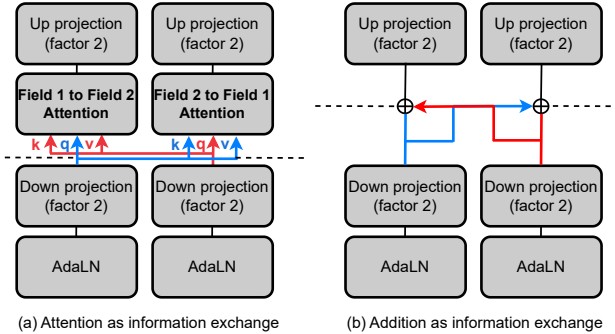

(a) Attention as information exchange    (b) Addition as information exchange

Figure 8: The information exchange mechanisms evaluated for this work, these include the information exchange with (a) cross-attention and (b) addition.

experiments, the variational encoding does not lead to significant improvements, and the embedded space error follows the real error (Refer to Appendix D). Therefore, we choose to proceed with pointwise encoding, as it consistently results in better reconstruction accuracy in the studied cases.

The hyperparameters associated with this autoencoder include the number of patches, the MLP dimensions and architecture, the number of heads in the attention mechanism, the number of attention layers, and the embedding dimension. The attention block is configured similarly to the original ViT, with exact specifications provided in Appendix G. The number of patches is primarily determined by the number of cells and the mesh dimensions. In this work, we use a constant number of 64 patches, arranged as 8 patches along the x-direction and 8 patches along the y-direction. Consequently, the number of cells per patch depends on both the total number of cells and the mesh density. Table 3 summarizes this information.

Table 3: Patch and cell information after padding for the cylinder flow and the multiphase flow experiments.

| Case | patch in x | patch in y | $C_p$ | $D$ |
|---|---|---|---|---|
| Cylinder flow | 8 | 8 | 232 | 16 |
| Multiphase flow | 8 | 8 | 156 | 32 |

## B    Temporal transformer detail

The transformer used to capture temporal dependencies is designed with a decoder-only architecture, incorporating the SEA and TIPI modules. In this work, we develop and evaluate two variations of information exchange methods, namely exchange by addition and exchange via cross-attention, as shown in Figure 8. Both methods operate within an information bottleneck, where the data is down-projected to retain only the relevant information required for exchange. To ensure improved gradient flow and prevent information loss due to this bottleneck, we introduce a residual connection from the output of the multi-head self-attention (MHSA) to the flow outside the SEA module. The down-projection within the bottleneck is set to a factor of 2, following the approach in [Lee et al., 2024]. The architecture referred to as the basic transformer does not incorporate any connections or information exchange between the decoders assigned to different variables.

Additionally, the TIPI module is incorporated as a direct conditioning mechanism, while AdaLN serves as an indirect conditioning technique. In the final design, the primary TIPI module utilizes the addition-based approach, as shown in Figure 9. However, we also explored injection via attention, which is commonly used as a conditioning technique in physics-based models.

The adaptive layer normalization (AdaLN) technique, used as the indirect conditioning method, is adopted from [Peebles and Xie, 2023]. In this approach, conditioning is applied by replacing the standard layer normalization with scale and shift parameters derived from the time-invariant parameter $\Theta$. To remain consistent with [Peebles and Xie, 2023], we use an MLP to compute the

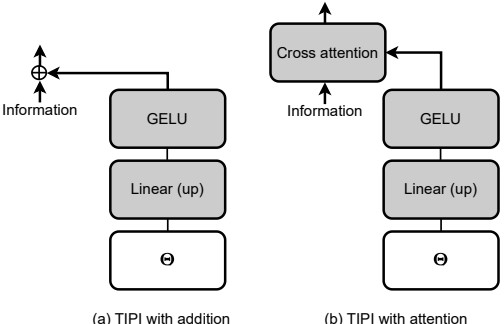

(a) TIPI with addition    (b) TIPI with attention

Figure 9: The information exchange mechanisms evaluated for this work, these include the information exchange with (a) cross-attention and (b) addition.

scale and shift values. First, $\Theta$ is up-projected to the model dimension, followed by the application of the SiLU activation function. A subsequent layer is then used to obtain the final scale and shift parameters.

An schematic of the AdaLN block is presented in Figure 10.

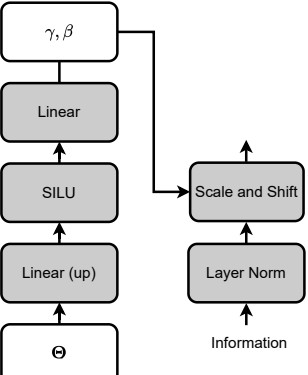

Figure 10: The block of adaptive layer norm adopted in the temporal model.

For the temporal model, the data is first structured similarly to the section A. After the encoding process, we obtain a data structure of [B*T, P, D] for each field group. To process different trajectories separately and avoid mixing, we separate the time from the batch and collapse the patch in the encodings. Hence, the final shape processed by the temporal model is [B, T, P*D], where B represents different trajectories, which we batch from and learn the temporal evolution.

## C    Training procedures

**Autoencoder**: The autoencoder was trained wtih the objective of the reconstruction error with L2 loss:

$$\mathcal{L} = \mathbb{E}_{\mathbf{X} \sim \mathcal{D}} \left[ \|\mathbf{X} - \hat{\mathbf{X}}\|_2^2 \right] \tag{6}$$

Where $\hat{\mathbf{X}}$ represents the model's output, and the expectation is done with respect to the data drawn from the batch.

For a consistent comparison with the literature we provide the errors using relative mean squared error where we refer to this as relative reconstruction error, and relative rollout error for the autoencoder

and temporal models respectively. The relative rollout MSE is formulated as:

$$\text{RelMSE} = \mathbb{E}_{\mathbf{X} \sim \mathcal{D}} \left[ \frac{\sum_{i=1}^{N} \|\mathbf{X}_i - \hat{\mathbf{X}}_i\|_2^2}{\sum_{i=1}^{N} \|\hat{\mathbf{X}}_i\|_2^2} \right] \tag{7}$$

Here the summation is done on the cell dimension.

**Temporal model**: The temporal model is trained using L2 loss in a teacher forcing manner, where the model generates the encoded field for the subsequent time step after observing the full trajectory. To accurately assess the autoregressive performance of the model, we conduct an autoregressive evaluation every 250 epochs, allowing us to track the learning history for the required autoregressive task. This evaluation includes monitoring both the encoded and decoded errors. We observe that, although the magnitude of variations in error is not perfectly aligned, the trends in the embedding space and decoded space errors follow a similar trajectory, peaking simultaneously. The following figure provides an example of the autoregressive learning curves for both the embedded space and the decoded space, using the ViT mesh autoencoder described in earlier sections.

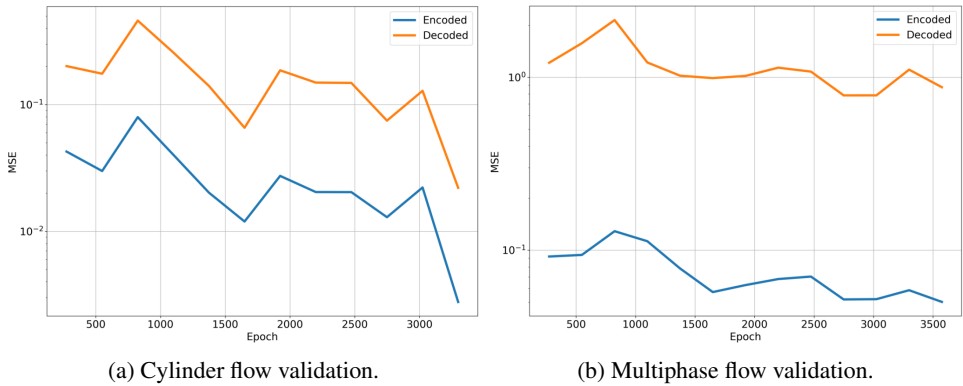

(a) Cylinder flow validation.  (b) Multiphase flow validation.

Figure 11: Comparison of the autoregressive evaluation in embedding space and after decoding where we observe they follow the same trend.

Provided curves in Figure 11 demonstrate the dynamic of the joint ViT-autoencoder and the temporal model. Informed by these curves, a non-variational encoding approach is adopted, and we assign 0 weights to the KL term used for the regularization of the latent space.

# D   Ablation study

Ablation of the information exchange method is performed using a vanilla architecture. The architecture incorporates layer normalization, a common practice in transformer models within the physics domain. We also initialize with absolute positional embeddings. Following our exploration of the vanilla transformer with the information exchange module, we extend the ablation studies to include the components of the main architecture. Specifically, we show that adding TIPI and AdaLN as conditioning mechanisms further reduces error and eliminates the need for an explicit attention mechanism for conditioning.

Table 4: Effect of different information exchange modes and no explicit exchange mode on the RelMSE. Reported errors are multiplied by $10^3$

| mode | CF (u) | CF (v) | CF (p) | MP (u) | MP (v) | MP ($\alpha$) | **Avg** |
|---|---|---|---|---|---|---|---|
| No exchange | 10 | 626 | 15.3 | 1043 | 1352 | 136.8 | 530.5 |
| Mixture-16 | 8.4 | 550.8 | 10.1 | - | - | - | - |
| Mixture-32 | 8.6 | 579.2 | 10.7 | 984.3 | 1235.8 | 258.9 | 512.9 |
| Mixture-64 | - | - | - | 938.5 | 1285.5 | 127 | - |
| Addition | 4.9 | 265.1 | 6.2 | **874.3** | **1168** | 164.5 | 413.83 |
| SEA | **4.1** | **206.5** | **5.4** | 904.1 | 1213.1 | **116.9** | **408.35** |

Table 5: Effect of our primary conditioning mechanism TIPI on the relative MSE. Reported errors are multiplied by $10^3$.

| TIPI model | CF | MP |
|---|---|---|
| No conditioning | 158.63 | 780 |
| Attention | **80** | 755.5 |
| Early-TIPI | 102.5 | **692.9** |
| Late-TIPI | 110 | 744.6 |

To demonstrate the effect of the TIPI module, we investigate how different types of conditioning impact the error. This information is summarized in Table 5, where "no conditioning" refers to the absence of TIPI, and "attention" indicates conditioning through attention mechanisms. Additionally, we evaluate whether the optimal placement of the TIPI module is before or after the SEA module.

It is evident that all three types of conditioning significantly reduce the error across both datasets. Furthermore, our secondary conditioning mechanism, AdaLN, reduces the errors even further, resulting in the lowest recorded values. We compare the effect of using AdaLN versus not using AdaLN, along with the best errors obtained from Table 5, as shown in Table 6. The addition of the second conditioning mechanism yields similar results regardless of TIPI placement, effectively removing TIPI placement as a critical model parameter.

The effect of RoPE embedding has been widely investigated in the literature, and thus, we do not explore this any further. However, in our experiments, the incorporation of RoPE significantly improved convergence within this specific domain.

Table 6: Effect of our secondary conditioning mechanism on with inclusion of the standard TIPI module presented in this work.

| AdaLN | CF | MP |
|---|---|---|
| ✗ | 80 | 692.9 |
| ✓ | **38.7** | **685.9** |

# E    Theoretical error

To gain deeper insight into the problem and the sources of error, we explore the errors addressed by our model and framework. In this analysis, we disregard the spatial and temporal discretization errors originating from the numerical solver and instead focus on the potential errors introduced by the autoregressive models grounded in physical simulations.

The coupling of variables in PDEs is a common occurrence. Many physical systems are governed by sets of linear or nonlinear coupled PDEs. Examples include Maxwell's equations (describing electromagnetism), the Ginzburg-Landau equation (used in superconductivity and phase transitions), the Einstein field equations (in general relativity), and the Navier-Stokes equations (governing fluid dynamics).

A common approach to modeling these equations involves embedding the variables into a shared latent space, where the system is resolved temporally in an autoregressive manner. For long trajectories, an autoregressive approach becomes essential, even for temporally-aware diffusion models. Consequently, it is critical to ensure that the embedded space closely resembles the real physical fields to maintain accuracy throughout the prediction process.

Assume $\psi^1$, and $\psi^2$ are two non-equal coupled field variables governed by the following advection equation, where $f$ and $g$ are any functions enforcing the coupling:

$$
\begin{aligned}
\frac{\partial \psi^1}{\partial t} + c_1 \frac{\partial \psi^1}{\partial x} &= f(\psi^2), \\
\frac{\partial \psi^2}{\partial t} + c_2 \frac{\partial \psi^2}{\partial x} &= g(\psi^1),
\end{aligned}
\tag{8}
$$

Say $\Psi$ is a variable that includes both $\psi^1$ and $\psi^2$ then the entropy of $\Psi$ follows the following condition:

$$H(\Psi) > \max\{H(\psi^1), H(\psi^2)\} \tag{9}$$

For the rate-distortion trade-off, this implies that, at the same rate, we can achieve lower field-wise distortion when the embedding is performed separately. In the case of two fields, achieving the same distortion requires doubling the rate to embed them together. This results in a higher-dimensional embedding space, which introduces two significant challenges for the temporal model. First, transformers scale quadratically with the embedding dimension, leading to greater computational complexity when the dimension increases, as opposed to using separate decoders for each field. Second, in higher dimensions, our temporal model begins to suffer from sparsity, resulting in poor generalization. Let the distortion error be denoted as $\epsilon_D$. Moreover, embedding the physical fields together may result in an uneven distribution of information from the fields, potentially weakening the coupling described in Equation 8.

We can now demonstrate how any form of information exchange theoretically leads to lower error when separate embedding spaces are created. Based on the definition of mutual information, we have:

$$I(\psi^1; \psi^2) = H(\psi^1) - H(\psi^1 \mid \psi^2), \tag{10}$$

where $I(\psi^1; \psi^2)$ represents the mutual information between $\psi^1$ and $\psi^2$, and $H(\psi^1)$ is the entropy of $\psi^1$. If $f(\psi^2)$ is non-zero, then from the PDE, we know that $I(\psi^1; \psi^2)$ is non-zero as well. This implies that the entropy of $\psi^1$, conditioned on $\psi^2$, is lower than the entropy of $\psi^1$ alone. A lack of such conditioning in the model introduces an additional error term due to increased uncertainty, which we refer to as the coupling error $\epsilon_C$. Our SEA module addresses this error by establishing a channel for information exchange between the fields.

To summarise the errors addressed in our work:

- Distortion error introduced the from joint embedding of the variables $\epsilon_D$.
- Coupling error introduced by lack of information exchange or unjust compression of information due to joint embedding $\epsilon_C$.

## F  Dataset information

The dataset used in this work consists of two fluid mechanics simulations motivated by physical phenomena: flow around a cylinder and the mixing of immiscible fluids. The governing equations for these problems are the conservation of mass (continuity equation) and the conservation of linear momentum (Navier-Stokes equation), which were numerically solved using the finite volume method implemented in the open-source software OpenFOAM. For the multiphase flow simulation, an additional advection-diffusion equation was solved to track the volume fraction of the phases. The results of these simulations were labeled and used as the ground truth in our experiments. The continuity and momentum equations for incompressible fluid flow are given by:

$$\nabla \cdot \boldsymbol{u} = 0 \tag{11}$$

$$\frac{\partial \boldsymbol{u}}{\partial t} + (\boldsymbol{u} \cdot \nabla)\,\boldsymbol{u} = -\frac{\nabla p}{\rho} + \nu \nabla^2 \boldsymbol{u} + \boldsymbol{g} \tag{12}$$

where $\nu$ is the kinematic viscosity, $\rho$ is the density, $p$ is the pressure, $\boldsymbol{u}$ is the velocity vector, and $\boldsymbol{g} = 9.81$ m/s$^2$ is the acceleration due to gravity.

The simulation of flow around a 2D cylinder was conducted using *icoFoam*, a transient incompressible solver for laminar flows in OpenFOAM. To ensure comparability with the literature, we generated 70 trajectories at different Reynolds numbers, with 60% used for training and 20% for validation. The input flow variables for this test case are the velocity components in the $x$- and $y$-directions $(u, v)$ and pressure $(p)$.

In the second test, the mixing of two fluids in the liquid phase with different densities was simulated using the *twoLiquidMixingFoam* solver. The flow was modeled as transient, incompressible, and

isothermal, with the dynamics of the mixture being the primary focus of this test case. The model input parameters included the velocity components and the volume fraction, which separates the fluids and ranges between 0 and 1. We generated 40 trajectories for this case, with a similar ratio for training, validation, and testing as in the cylinder flow case.

## G    Final configuration and hyperparameters

Table 7 details the configurations for the ViT-based mesh autoencoder and the temporal model, including key hyperparameters such as the number of patches and TIPI-specific settings. These include scaling mechanisms (MLP, Gaussian Fourier projection, linear) and injection methods (addition, attention). The bottleneck down-projection, adapted from Lee et al. [2024], is introduced as an information exchange-specific hyperparameter.

Table 7: The configurations and parameters for the ViT-based mesh autoencoder, temporal transformer architecture, and training parameters. $\alpha$ represents the volume of fluid for multiphase flow (Refer to Appendix F).

| ViT-based Mesh Autoencoder | | |
|---|---|---|
| Model parameters | Cylinder flow | Multiphase flow |
| Number of layers | 12 | 12 |
| Number of heads | 8 | 8 |
| MLP scale ratio | 4 | 4 |
| Embedding size | 16 | 32 |
| Dropout | 0 | 0 |
| Variational | ✗ | ✗ |
| Patches in x | 8 | 8 |
| Patches in y | 8 | 8 |
| Field groups | $\{u, v\}, \{p\}$ | $\{u, v\}, \{\alpha\}$ |
| **Temporal Transformer Architecture** | | |
| Model parameters | Cylinder flow | Multiphase flow |
| Number of layers | 1 | 1 |
| Number of heads | 8 | 8 |
| MLP scale ratio | 8 | 8 |
| Model size | 1024 | 2048 |
| Dropout | 0.3 | 0 |
| Down projection | 2 | 2 |
| TIPI scale mode | MLP | MLP |
| TIPI injection mode | add | add |
| **Training Parameters** | | |
| **Autoencoder** | | |
| Optimizer | AdamW | AdamW |
| Learning rate | 1e-4 | 1e-4 |
| Weight decay | 1e-5 | 1e-5 |
| Optimizer momentum $(\beta_1, \beta_2)$ | (0.9, 0.999) | (0.9, 0.999) |
| Batch size (snapshot) | 128 | 128 |
| Epochs | 600 | 1000 |
| **Temporal Model** | | |
| Optimizer | AdamW | AdamW |
| Learning rate | 1e-4 | 8e-5 |
| Weight decay | 1e-5 | 1e-5 |
| Optimizer momentum $(\beta_1, \beta_2)$ | (0.9, 0.999) | (0.9, 0.999) |
| Batch size (trajectory) | 2 | 4 |
| Epochs | 3500 | 2500 |

## H    Additional results

Figures 12 and 13 preset contour of velocity components and pressure for the flow around the cylinder. Figures 14–16 show the contour of velocity components and volume fraction for the mixing two fluids test case.

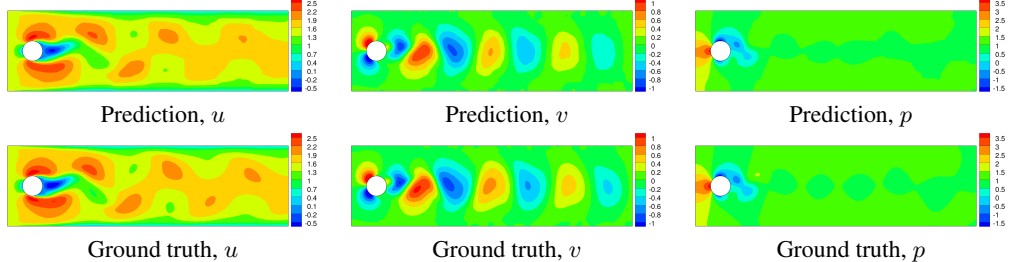

Figure 12: Contour maps of the generated fields at Re=800, and time step of 250 where Von Karman vortex street is formed.

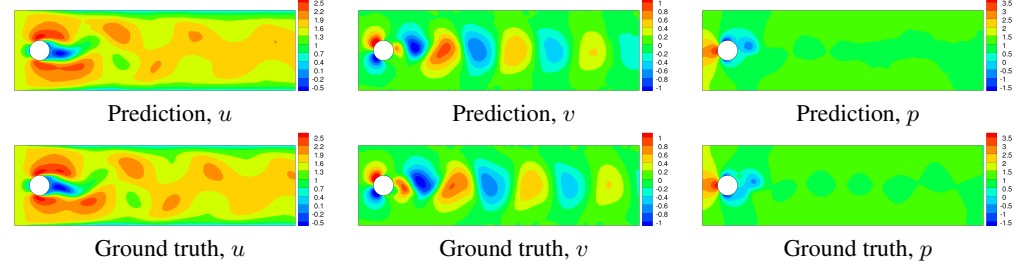

Figure 13: Contour maps of the generated fields at Re=850, and time step of 250 where Von Karman vortex street is formed.

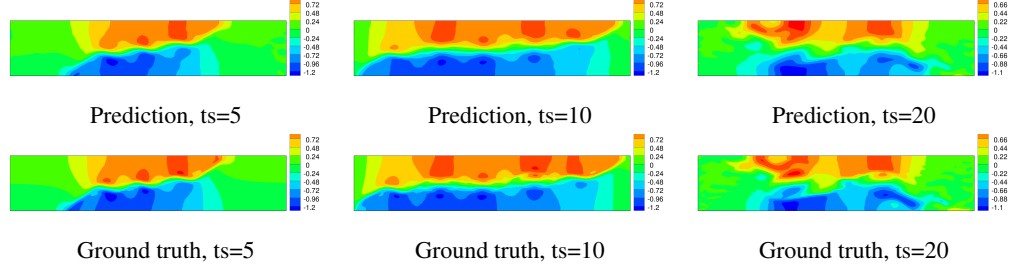

Figure 14: Comparison of the contour maps of the velocity component $u$, between the predictions and ground truth in the case of $\rho = 850$.

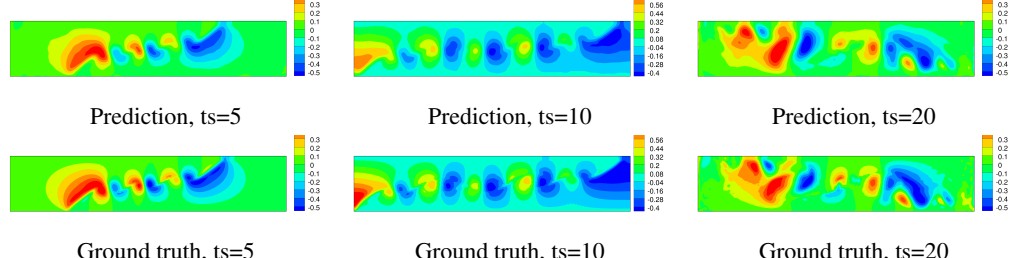

Figure 15: Comparison of the contour maps of the velocity component $v$, between the predictions and ground truth in the case of $\rho = 850$.

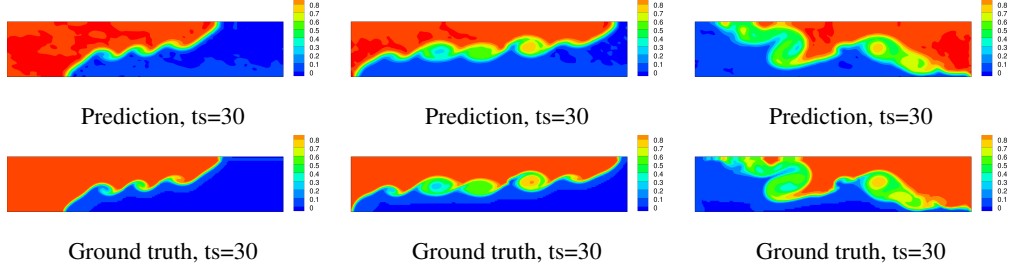

Figure 16: Comparison of the contour maps of the volume fraction ($\alpha$), and velocity components between the predictions and ground truth in the case of $\rho = 850$.

