# OpenReview forum: "SEA: State-Exchange Attention for High-Fidelity Physics Based Transformers"
_NeurIPS.cc/2024/Conference — NeurIPS 2024 poster_

### Official Review · Reviewer_Yocw · 2024-06-17

**Soundness:** 3
**Presentation:** 4
**Contribution:** 4
**Rating:** 8
**Confidence:** 4

**Summary:**

This paper presents a novel neural network architecture (SEA) for solving partial differential equations (PDEs) in physical problems. This architecture effectively utilizes information exchange between multiple fields and the conservation quantities of physical systems to correct model predictions, achieving smaller errors and improved generalization. This has significant value for industrial applications, as it can greatly reduce the time cost for solving specific problems.

**Strengths:**

**Originality:** The authors propose a novel neural network architecture (SEA) that effectively utilizes information exchange between multiple fields while cleverly incorporating physical conservation quantities. This maximizes the use of the physical system's symmetry to reduce errors.

**Clarity:** The paper clearly defines concepts and provides a clear and explicit explanation of the motivation for proposing this architecture. The presentation of experimental results is detailed, and the overall model structure is presented in an intuitive and clear manner.

**Significance:** The model in this paper offers valuable practical insights into integrating neural networks into PDE solvers, providing a fresh perspective in this field. This has important implications for enhancing productivity in the industrial sector.

**Quality:** The language of the paper is rigorous and clear, with appropriate citations of previous work. The overall approach is straightforward and easy to understand. The experimental results include comparisons with multiple similar models for reasonable and effective evaluation.

**Weaknesses:**

Further exploration of the model's generalization ability is needed, such as evaluating model errors under more complex boundary conditions. Additionally, presenting the training and inference costs of the model would provide a better assessment of its potential for industrial applications.

**Questions:**

See cooments

**Limitations:**

yes

---

> ### Author Rebuttal · Authors · 2024-08-06
>
> ***
> **W1. Further exploration of the model's generalization ability is needed, such as evaluating model errors under more complex boundary conditions. Additionally, presenting the training and inference costs of the model would provide a better assessment of its potential for industrial applications.**
>
> Our future work will focus on addressing the scalability of the architecture and evaluating the model on a wider range of scenarios. We plan to conduct further evaluations using the PDEBench dataset and other datasets from different domains. Details on costs and time complexity are discussed in the global rebuttal and provided in the attached PDF.

---

> > ### Comment · Reviewer_Yocw · 2024-08-14
> >
> > Thanks for your responses. They help resolve my questions.

---

### Official Review · Reviewer_M5vu · 2024-07-12

**Soundness:** 4
**Presentation:** 4
**Contribution:** 3
**Rating:** 7
**Confidence:** 3

**Summary:**

The paper introduces the State-Exchange Attention (SEA) module, a new cross-attention module that enables information exchange between state variables in physics-domain transformer modules. The authors evaluate the performance of this module in both single-phase and multi-phase 2D fluid settings.

**Strengths:**

1. Novelty: The State-Exchange Attention architecture represents an interesting and new contribution in architecture for physics-domain transformer models. This module enables multi-directional information exchange between state variables, and as far as I know is the first of its kind for physics-domain transformer architectures.
2. Performance: The SEA module significantly reduces rollout error accumulation, which is a major issue in current sequential models. The paper reports substantial improvements over competitive baseline models in both a single-phase and multi-phase context.
3. Strong Evaluation: The SEA-integrated model is well evaluated across different computational fluid dynamics (CFD) cases, showing consistent performance improvements. This includes detailed experiments on both single-phase and multiphase flows.

**Weaknesses:**

1. Scalability Concerns: The current architecture is relatively small, using a transformer with only 1 layer and 8 attention heads. I am curious how the model will scale at larger architecture sizes. Moreover, as the authors themselves note, there are also concerns about how the model will scale with larger number of state variables.
2. Diversity of experiments: The authors evaluate performance on one instance of single-phase and one instance of multi-phase flow, both in two dimensions. How does the model fair in 1D and 3D settings? How does it fair in a wider variety of fluid mechanics problems (like those presented in PDEBench/PDEArena)?

Minor comments:
- It seems like the general gist of paragraphs on lines 179-186 and lines 187-192 are almost identical. Was this paragraph repeated by accident?

**Questions:**

1. Are there ablation studies available for single-phase flow (as is sort of provided for multi-phase flow). In other words, how does a basic transformer fair in this setting?
2. How resource-intensive is training this model vs. traditional transformer architectures? Some indication of this would be very helpful.

**Limitations:**

See weaknesses.

---

> ### Author Rebuttal · Authors · 2024-08-06
>
> ***
> **W1. Scalability Concerns: The current architecture is relatively small, using a transformer with only 1 layer and 8 attention heads. I am curious how the model will scale at larger architecture sizes. Moreover, as the authors themselves note, there are also concerns about how the model will scale with larger number of state variables.**
>
> As dictated by the available data and complexity trade-off, increasing the complexity of the model would not be ideal in either case. We carefully studied the architectural aspects and converged on the same architecture as the transformer based works we compared our results to [1,2]. We also provide the our model dimension study in the provided PDF..
> ***
> **W2. Diversity of experiments: The authors evaluate performance on one instance of single-phase and one instance of multi-phase flow, both in two dimensions. How does the model fair in 1D and 3D settings? How does it fair in a wider variety of fluid mechanics problems (like those presented in PDEBench/PDEArena)?**
>
> Our model is specifically designed to address the coupling between state variables governed by different PDEs. In a 1D scenario, reducing the model will yield the same results as those found in the literature, as there is only one state variable and information exchange becomes redundant (here, we focus on the temporal model and not the ViT encoder-decoder). The 2D problems studied, such as the 2D Navier-Stokes and 2D Navier-Stokes with volume of fluid PDE, were chosen due to the strong coupling between the variables. We avoided 3D cases in this study for clearer demonstrations, however the model is by no means limited to 2D.
> ***
> **W3. It seems like the general gist of paragraphs on lines 179-186 and lines 187-192 are almost identical. Was this paragraph repeated by accident?**
>
> This is indeed an editting problem, we appreciate your detailed revision and pointing this out.
> ***
> **Q1. Are there ablation studies available for single-phase flow (as is sort of provided for multi-phase flow). In other words, how does a basic transformer fair in this setting?**
>
> The ablation studies are presented in an additional table within the provided PDF and the supplementary materials. This table demonstrates the model's performance under different information exchange modes: addition of information for exchange, SEA attention mechanism for exchange, and no information exchange. Additionally, this table includes the case where state variables share a mutual latent space, and no explicit information exchange is defined.
>  ***
> **Q2. How resource-intensive is training this model vs. traditional transformer architectures? Some indication of this would be very helpful.**
>
> We have added the information regarding the training time to the global rebuttal, and further demonstrated the inference time for the full trajectories of studied cases with respect to the model dimension in the provided PDF.
>
> ***
> # References
> [1] Luning Sun, Xu Han, Han Gao, Jian-Xun Wang, and Liping Liu. Unifying predictions of deter-ministic and stochastic physics in mesh-reduced space with sequential flow generative model. In A. Oh, T. Naumann, A. Globerson, K. Saenko, M. Hardt, and S. Levine, editors, Advances in Neural Information Processing Systems, volume 36, pages 60636–60660. Curran Associates, Inc., 2023.
>
> [2] Xu Han, Han Gao, Tobias Pfaff, Jian-Xun Wang, and Li-Ping Liu. Predicting physics in mesh-reduced space with temporal attention. arXiv preprint arXiv:2201.09113, 2022.

---

> > ### Comment · Reviewer_M5vu · 2024-08-10
> >
> > Thank you for your thoughtful response and for the additional details in the provided PDF. I am satisfied with the authors comments and additional details. I will keep my score.

---

### Official Review · Reviewer_NXYL · 2024-07-20

**Soundness:** 3
**Presentation:** 3
**Contribution:** 3
**Rating:** 7
**Confidence:** 3

**Summary:**

The authors present a novel and interesting approach to physics-domain transformer models that exchanges information between state variables and demonstrates strong performance improvements relative to SOTA models on hard fluid dynamics problems. The paper is well written and the results compelling.

**Strengths:**

I find the multidimensional information exchange architecture employed by the authors to be interesting, the improvement over existing models to be significant on challenging problems, and the paper and figures to be quite well done. I would like to see more thorough evaluation of the models, perhaps including application to a non-fluid domain or to a problem with a few more underlying state variables, and it is highly desirable that code be made available with the paper.

**Weaknesses:**

#### Major concerns:

1. Perhaps I missed this somewhere, but will code be made available with this paper? It seems too much to require readers to re-implement this architecture to use it.

2. The supplement is helpful but could contain further experimental details, further details on the layers (and their trainability), how projections are performed, and more complete explanation of results and metrics.

3. If training the models takes just 15-20 minutes, it would be nice to see the effects of the embedding dimensionality and simulation parameters explored more thoroughly.

4. The authors point out that a limitation might be scaling to a large number of state variables. Could they present an example on a problem with more than two state variables so that the "multidirectional" component is more than bidirectional, and perhaps add some thoughts on how this might scale?

#### Minor concerns:
There are a number of places the notation could be clarified and some typos, for example:
By line:
- 126: argmax optimization variable is not specified (presumably z_1:T)
- 159: the X variable is never defined, and may be superfluous here... maybe just use z_i^k?
- 166: T should be italic?
- A number of variables throughout seem to be used without introduction.

**Questions:**

- How does the padding work in the case of graphs with highly variable node density (for example on an adaptive mesh with much denser representation near the cylinder as is common in practice)?

**Limitations:**

The authors have touched on potential limitations related to number of state variables but could discuss this in more detail, as many problems of interest contain more than two state variables.

---

> ### Author Rebuttal · Authors · 2024-08-06
>
> ***
> **W1 and W2: Perhaps I missed this somewhere, but will code be made available with this paper? It seems too much to require readers to re-implement this architecture to use it.**
>
> The code is prepared and will be released with the paper. We will also add more information in the implementation of supplementary materials for better reproducibility.
> ***
> **W3: If training the models takes just 15-20 minutes, it would be nice to see the effects of the embedding dimensionality and simulation parameters explored more thoroughly.**
>
> The optimal embedding dimensions were determined for both cases by experimenting. It was found that lower embedding dimensions made reconstruction difficult and amplified errors, while higher dimensions required more computation and led to model overfitting. A detailed study, included in the provided PDF (Figure 1) and will be added to the supplementary material, to further clarify the chosen dimensions.
>
> ***
> **W4: The authors point out that a limitation might be scaling to a large number of state variables. Could they present an example on a problem with more than two state variables so that the "multidirectional" component is more than bidirectional, and perhaps add some thoughts on how this might scale?**
>
> One potential field in the physics and material sciences with a large number of state variables would be the phase field simulations to capture the mesoscale dynamics of the grains. These 3D problems tend to have one state variable per grain, resulting in M-coupled equations where M represents the number of variables. Another example would be the multiphase flows with more than two phases. Given the history of each variable is resolved with its corresponding transformer, this work becomes limited in these cases. Even though the number of variables does not theoretically limit the model itself the required computation increases quadratic with the number of variables. However, this is not different from other works in the literature where all the variables are in the same latent space, and one transformer resolves the dynamics of all variables. In those cases, the latent dimension must be increased when more variables are stored on the mesh, resulting in higher computation. Scalability is further discussed in the global response. We also intend to address the scalability in the future works and include a wider range of test cases in different domains to evaluate the performance of the model.
>
> ***
> **minor problems:**
> We appreciate the corrections. These problems were all corrected.
>
> ***
> **Q1. How does the padding work in the case of graphs with highly variable node density (for example on an adaptive mesh with much denser representation near the cylinder as is common in practice)?**
>
> In this study, all patches are uniformly sized, similar to the Vision Transformer (ViT). To ensure consistency, each patch is padded with zeros up to the maximum number of elements found in any patch within the domain. Prior to the mentioned padding, we introduce a small noise in the fields stored on the mesh such that there is no element with an exact value of 0 (In order to keep these elements active in the learning and separate them from the pads). Further, we have adopted a GeLU activation function during the downsampling of each patch, resulting in no contribution from the pads in the encoder-decoder pair. This treatment, by default, captures more information in the patches with high mesh density and lower information in those with a more sparse structure (Highly padded patches).

---

> > ### Comment · Reviewer_NXYL · 2024-08-10
> > **Thank you.**
> >
> > We are satisfied with the authors response and have increased our score accordingly.

---

### Official Review · Reviewer_Nand · 2024-07-21

**Soundness:** 3
**Presentation:** 4
**Contribution:** 3
**Rating:** 6
**Confidence:** 4

**Summary:**

The study proposes an approach for autoregressive spatiotemporal estimation of a dynamical system state, e.g. solution of time-dependent PDE. In particular, a Vision Transformer (VIT) model is adapted to solving PDEs where each state variable is tokenized similarly to tokenizing an image and a novel State-Exchange Attention (SEA) module allows interaction and fusion of these tokens through cross-filed attention. This enables state variables in the system represented by tokens to exchange information and capture the physical relationships and symmetries between fields. The approach is evaluated on the problem of flow past a cylinder benchmark showing improvement of the error against current neural networks approaches.

**Strengths:**

S1. The work presents a novel system based on vision transformer to solve PDEs in autoregressive way. The system is constructed in logical way and the SEA component developed for interaction of field variable is sensible and appears to be applicable for capturing such interactions.

S2. Evaluation of the model on 2D flow past a cylinder shows advantage of the model vs. existing approaches.

S3. The model seems to be generalizable to other PDEs and even to simulations of physical processes that do not have formulated equations.

**Weaknesses:**

W1. It is claimed that the rollout errors are effectively controlled by the SEA model. There's very little explanation/derivations of the source of these errors. It would be helpful to define these errors rigorously and also if possible to show (graphically or rigorously) how SEA model allows for better control of the error.

W2. Ablations of SEA model components and how their contribution to the overall accuracy of solution are missing. These could inform the critical parts of SEA and better characterize SEA.

W3. While generalization seems plausible, evaluation was performed on only one basic dataset from which introduces uncertainty in how the approach generalizes.

**Questions:**

Q1. What is the computational complexity (time, storage) of SEA and how does it compare to other approaches and PDE simulation?

Q2. Does SEA apply to high dimensional state variables, e.g. 3D flows or ND state variables? If yes, it would be important to include in the manuscript (appendix) practical steps of extending the model to higher dimensions.

**Limitations:**

The technical contributions of this work are subtle since ViT are well known as powerful models and the SEA component seems to be sensible and inherits structure from prior approaches. This could be considered as a limitation, however, on the other hand, the impact of this work lies in the application of solving dynamical equations and simulating dynamical processes and appears to be significant advancement. The ability to implement existing components in such a way that they will be effective is not a straightforward endeavor. Hence, ablation results, discussion of computational complexity and more explanation of the error would be vital in this work. Also, addressing Weaknesses and Questions above would improve clarity, generalization and evaluation of this promising work.

---

> ### Author Rebuttal · Authors · 2024-08-06
>
> ***
> **W1. It is claimed that the rollout errors are effectively controlled by the SEA model. There's very little explanation/derivations of the source of these errors. It would be helpful to define these errors rigorously and also if possible to show (graphically or rigorously) how SEA model allows for better control of the error.**
>
> The overall errors of this model with regards to the numerical results (Neglecting the discretization errors) consist of two main components, namely reconstruction error (Relevant to the encoder-decoder) and temporal error (Relevant to the temporal transformer). The SEA module explicitly addresses the temporal error accumulating during inference (Autoregressive generation) to form the total rollout error. Given a set of coupled partial differential equations with M-state variables, the optimal temporal model would estimate the exact solution as the discretized numerical model. It should be noted that a discretized equation for each variable contains all or a set of other state variables since these equations are coupled. Hence, embedding all the variables in the same embedding space with the sole objective of reconstruction loss may result in unjust information from each field, which would result in an additional error term regarding the coupling of different variables. Our work eliminates this source of error by creating different embedding spaces for different variables, and the SEA module mimics the coupling between different variables. A theoretical analysis has been prepared and will be attached to the supplementary materials.
> ***
> **W2. Ablations of SEA model components and how their contribution to the overall accuracy of solution are missing. These could inform the critical parts of SEA and better characterize SEA.**
>
> The ablation study is summarized in table 1 provided in the attached PDF. This table will also be added to the supplementary material for better understanding of the model. The included table contains the following information:
> 1. performance of the model with SEA module for information exchange
> 2. Addition of the states to other states for information exchange
> 3. No information exchange (variables solved with different transformers)
> 4. A single transformer architecture with a shared latent space for all variables (Common in literature).
> This table demonstrates that models utilizing modes of information exchange outperform those without information exchange or with a shared latent space.
> ***
> **W3. While generalization seems plausible, evaluation was performed on only one basic dataset from which introduces uncertainty in how the approach generalizes.**
>
> Cylinder flow and the multiphase cases here were chosen to demonstrate the model's performance in cases with different governing PDEs. In the multiphase case, in addition to the Navier stoke's, the volume of fluid PDE is solved coupled with the momentum. In this case, we focus on demonstrating how the SEA module creates this coupling between volume fraction and velocity. Given the clear decreasing trend of error on every variable, as shown in the ablation table in the provided PDF, and the identified sources of error (Explained in W1), we believe this model generalizes well to cases with different governing PDEs, initial conditions, or boundary conditions.
> ***
> **Q1. What is the computational complexity (time, storage) of SEA and how does it compare to other approaches and PDE simulation?**
>
> The time complexity of the encoder-decoder is similar to the transformer and is unaffected by the number of state variables. Similarly the storage is the same as the ViT base model. However, the temporal model with the SEA module has an additional quadratic term regarding the number of state variables. If we denote the number of state variables as M, the time complexity becomes O(M^2 * D * N^2) where D, and N are the model dimension and the input block size respectively.
> The current state-of-the-art studies resolving full trajectories use techniques with a transformer backbone, which is dominated by the self-attention term O(D*N^2) [1,2].
> Although the inference time remains orders of magnitude below the numerical simulations, the time complexity increases quadratically with respect to the state variables which is the main limitation of this module. More information on computational aspects are provided in the global section.
>
> **Q2. Does SEA apply to high dimensional state variables, e.g. 3D flows or ND state variables? If yes, it would be important to include in the manuscript (appendix) practical steps of extending the model to higher dimensions.**
>
> The model is capable of 3D flows, and in general n dimensional and multi-variate problems. The procedure for this will be added to the appendix and the the code will be released with the paper which includes a complete implementation for any specified number of variables.
> ***
> # References
> [1] Luning Sun, Xu Han, Han Gao, Jian-Xun Wang, and Liping Liu. Unifying predictions of deter-ministic and stochastic physics in mesh-reduced space with sequential flow generative model. In A. Oh, T. Naumann, A. Globerson, K. Saenko, M. Hardt, and S. Levine, editors, Advances in Neural Information Processing Systems, volume 36, pages 60636–60660. Curran Associates, Inc., 2023.
>
> [2] Xu Han, Han Gao, Tobias Pfaff, Jian-Xun Wang, and Li-Ping Liu. Predicting physics in mesh-reduced space with temporal attention. arXiv preprint arXiv:2201.09113, 2022.

---

> > ### Comment · Reviewer_Nand · 2024-08-13
> > **Authors Rebuttal Response**
> >
> > I'd like to thank the authors for providing an informative rebuttal and additional results in response to comments that I made in my review.  The authors clarified the questions and comments that I made and I would like to encourage the authors to include the results in the provided pdf and their discussion in the revision and also rigorous definition of the involved errors along with succinct explanation as the authors provide in the rebuttal to W1.

---

### Author Rebuttal · Authors · 2024-08-06

***
We would like to appreciate all referees for spending time giving great feedback, and instructions on how to improve the paper.

Here we would like to address two main points that were questioned by all the referees, first the limitation of the work and then the efficiency of the model.

***
# Limitation
We first briefly review the limitation and then suggest a potential way forward for the future works.
The SEA module effectively acting as a cross attention amongst the state variables is quadratic in time with respect to the number of state variables which limits the module extending to large number of variables. The time complexity of this model can be expressed as O(M^2 * D * N^2) , where M is the number of variables, D is the dimension of the model, and N is the sequence length. However, various strategies can potentially be studied in future work to mitigate this complexity. A key technique involves leveraging prior knowledge of the equations or the physical system to inform the cross-attention process. This prior knowledge can be utilized to adjust the cross-attention between variables, ensuring the method is applied only to strongly coupled variables (e.g., velocity and pressure). Another approach involves embedding variables of the same type together. For example, in the 3D Navier-Stokes equations, the three components of velocity would be embedded together, while source terms like pressure and other external or internal forces would be embedded together. In our future work, we will conduct a comprehensive investigation of these factors. Additionally, we will explore the implementation of this model within a probabilistic framework (Guiding the output based on the coupling between variables), leveraging its ability to effectively resolve the history of variables and their coupling and address uncertainty during inference.

***
# Computational aspects
Another point raised by all the reviewers pertains to the computational aspects of the model. We have recorded the training and inference times for both cases. In the paper, the reported training was conducted using teacher forcing on both the training and validation sets. Additionally, we present the training time for the scenario where validation is tracked in an autoregressive manner. While both approaches yield meaningful training, we include the autoregressive validation case because the model's ultimate task is to estimate full trajectories.
The training for all cases is done with a single A100 GPU. In cylinder flow we resolve 5600 time points, with 7697 meshing points, and 3 variables of u, v, and p stored on each point.
For the multiphase case we have a total of 7400 time steps and 8241 meshing points. Similar to the cylinder flow we have three variables here namely u, v, and alpha where alpha is the volume fraction capturing the phases and their interface. The following computational detail will enter the supplementary materials.

The training cost of each case is listed below:
1) Training with validation using teacher forcing
* Cylinder flow: 20-30 Minutes
* Multiphase flow: 60-70 Minutes*

2) Training with validation using autoregressive approach
* Cylinder flow: 45-60 Minutes
* Multiphase flow: 2 Hours

*A typo in the main manuscript reported the multiphase flow training time as 15 minutes. The correct training time is provided here and will be corrected in the paper.

Inference of full trajectory for all cases remains below 30 seconds a detailed plot of which with respect to model dimension is provided in the attached PDF.

***
# Provided PDF
In the provided PDF we have included a detailed ablation study which also contains results of the case with shared latent space between all variables (Table 1). We also evaluate the performance of the time invariant parameter injection module in table 2. Another critical parameter in training this model is the batch size. Increasing the batch size was observed to raise the error in autoregressive generation and potentially cause divergence. Relevant information on the batch size is provided in Table 2. To fully understand the impact of dimension choice, we included several plots illustrating the reconstruction error (MSE) of the ViT encoder-decoder, temporal error (relative MSE), and inference time. These plots, along with the provided training cost, informed our dimension selection.

The provided results in the PDF will be appended to the supplementary materials along with a theoretical analysis of errors to demonstrate which component of error is being addressed by our model.

---

### Decision · Program_Chairs · 2024-09-25

**Decision:**

Accept (poster)

**Comment:**

Reviewers appreciate how the paper adapts vision transformers to PDEs with a "state exchange attention (SEA)" mechanism and a "time invariant parameter injection (TIPI)" component. I personally find the paper to be frustratingly lacking in key details, and I strongly encourage the authors to address these concerns in the camera ready version.

1. The authors do a decent job of motivating what they want to accomplish with SEA, but the mathematical description in eq. (3) doesn't make sense. Index $j$ appears on the right-hand side but not the left, and there's no specification of how results are reduced over $j$.
2. The so-called "time invariant parameter injection (TIPI)" component is not well motivated, and again the mathematical description in eq. (5) doesn't make sense: a mysterious "information injector" $f_{IJ}(\mu)$ is introduced without explanation. The variable $\mu$ is never even defined!
3. The captions for Figure 1 and Figure 2 are insufficient to describe what is being shown. The text sometimes refers to Figure 5 when it is meant to refer to Figure 3. The tick labels are illegible in Figures 4-6.
4. Table 1 presents the authors' method in bold even when it does not achieve the lowest error (specifically for $u$).
5. The authors use \citet when they should use \citep.
6. The paper needs to be edited for spelling mistakes, grammatical errors, and redundant phrases. For example, "We can formally formulate this problem..." (line 125) and "To process and prepare the spatial information..." (line 130).